# Extended Infusion of Meropenem in Neonatal Sepsis: A Historical Cohort Study

**DOI:** 10.3390/antibiotics11030341

**Published:** 2022-03-04

**Authors:** Guangna Cao, Pengxiang Zhou, Hua Zhang, Bangkai Sun, Xiaomei Tong, Yan Xing

**Affiliations:** 1Department of Pediatrics, Peking University Third Hospital, Beijing 100191, China; guangnacao@bjmu.edu.cn; 2Department of Pharmacy, Peking University Third Hospital, Beijing 100191, China; pxzhou0427@bjmu.edu.cn; 3Peking University Health Science Center, Institute for Drug Evaluation, Beijing 100191, China; 4Research Center of Clinical Epidemiology, Peking University Third Hospital, Beijing 100191, China; zhanghua@bjmu.edu.cn; 5Information Management and Big Data Center, Peking University Third Hospital, Beijing 100191, China; bangkaisun@bjmu.edu.cn

**Keywords:** meropenem, extended infusion, short-term infusion, neonatal sepsis, historical cohort study

## Abstract

This single-center historical cohort study investigated the effectiveness and safety of extended infusion (EI) compared with short-term infusion (STI) of meropenem in neonatal sepsis. Patient electronic health records from Peking University Third Hospital (1 December 2011–1 April 2021) were screened. Neonates diagnosed with sepsis and treated with meropenem in the neonatal intensive care unit were included (256 patients) as STI (0.5 h, 129 patients) and EI (2–3 h, 127 patients) groups. Three-day clinical effectiveness and three-day microbial clearance were considered the main outcomes. Univariate and multivariate analyses were performed. Baseline characteristics were similar in both groups. EI of meropenem was associated with a significantly higher 3-day clinical effectiveness rate (0.335 (0.180, 0.623), *p* = 0.001) and 3-day microbial clearance (4.127 (1.235, 13.784), *p* = 0.021) than STI, with comparable safety. Subgroup analyses showed that neonates with very low birth weight benefited from EI in terms of 3-day clinical effectiveness rate (75.6% versus 56.6%, *p* = 0.007), with no significant difference in the 3-day clinical effectiveness (85.1% versus 78.3%, *p* = 0.325) and microbial clearance (6% versus 5%, *p* > 0.999) rates between 3 h and 2 h infusions. Thus, EI of meropenem may be associated with better effectiveness and comparable safety in treating neonatal sepsis than STI. Nonetheless, historically analyzed safety evaluation might be biased, and these findings need confirmation in randomized controlled trials of larger sample sizes.

## 1. Introduction

Neonatal sepsis, a systemic inflammatory response syndrome caused by bacterial, viral, or fungal infections, was reported to be the third leading cause of neonatal death worldwide, with mortality between 11% and 19% [1,2]. A systematic review suggested that approximately 3 million neonates develop sepsis every year, with a global mortality rate of 19%, with Gram-negative bacilli being the main causative agent [3]. Severe neonatal sepsis needs to be treated with carbapenems, among which meropenem, the most widely used agent of this group, exhibits time-dependent bactericidal activity [4]. Nevertheless, with the widespread use of meropenem in neonatal wards, there is recognition of increasing carbapenem resistance and virulence, which are considerable challenges to antibiotic management. When dealing with a virulent resistant strain, or attaining no-therapeutic success with maximum doses, extended infusion (EI, 2 to 3 h) of meropenem (20 mg/kg/dose) compared with short-term infusion (STI, 0.5 h) is considered to achieve better pharmacokinetic (PK) and pharmacodynamic (PD) targets [5]. To date, the optimization of EI of meropenem has been investigated using PK or population PK [6] modeling, and trials based on small sample sizes [7]. However, there are no guidelines or clinical studies based on large sample sizes that provide definite recommendations on the application of EI of meropenem in treating neonatal sepsis. Therefore, here, we conducted a historical cohort study to investigate the effectiveness and safety of EI compared with STI of meropenem in neonatal sepsis based on real-world data.

## 2. Results

We initially identified 653 neonates that had been prescribed meropenem from their medical records, 397 of which were excluded since they were not diagnosed with neonatal sepsis, had comorbidities of Gram-positive cocci sepsis or purulent meningitis, or they were treated for less than 3 days (Figure 1). Thus, 256 neonates were divided into the EI (127 patients) and STI (129 patients) groups for analysis, which was larger than 123, the estimated sample size (see Section 4.1. Study Sample), so the results of this study could be considered credible. There was no significant difference between the baseline characteristics of the two groups (Table 1).

### 2.1. EI of Meropenem in Effectiveness Outcomes

Univariate analyses showed that neonates in the EI group experienced a significantly higher 3-day clinical effectiveness rate (81.9% versus 59.7%, *p* < 0.001) and 3-day microbial clearance (94.5% versus 85.3%, *p* = 0.015) than those in the STI group. However, there was no difference in the 3-day C-reactive protein (CRP) recovery rate (*p* = 0.141) and the 3-day white blood cell (WBC) recovery rate (*p* = 0.105) between the groups (Table 2).

Results of multivariate analyses showed that EI of meropenem was associated with a higher 3-day clinical effectiveness rate (*p* = 0.001) and 3-day microbial clearance (*p* = 0.021) than STI. In addition, lower gestational age (*p* = 0.010), hypotension (*p* = 0.020), extended capillary refill time (CRT; *p* = 0.018), vomiting (*p* = 0.037), and scleroderma (*p* < 0.001) were considered as predictive factors for 3-day microbial clearance (Table 3).

### 2.2. EI of Meropenem in Safety Outcomes

Overall, drug-related adverse events (AEs) in all neonates were mild, and there was no case of treatment discontinuation due to serious AEs. Furthermore, among the results of renal tests, there was no clinical and statistical difference in 3-day blood urea nitrogen (BUN; *p* = 0.383), creatinine (*p* = 0.160), and alanine transaminase (ALT; *p* = 0.724) abnormalities between the EI and STI groups (Table 4).

### 2.3. Subgroup Analysis

#### 2.3.1. EI of Meropenem in Very Low Birth Weight Infants

We conducted a subgroup analysis in 185 very low birth weight (VLBW) infants, including 86 patients in the EI group and 99 patients in the STI group; their baseline characteristics were not significantly different (Appendix A). Univariate analyses showed that neonates with VLBW benefitted more from EI of meropenem with regard to the 3-day clinical effectiveness rate (75.6% versus 56.6%, *p* = 0.007) and the 3-day microbial clearance rate (94.2% versus 84.8%, *p* = 0.041) than those in the STI group. There was no statistical difference in 3-day CRP change (*p* = 0.223) and 3-day WBC recovery rate (*p* = 0.809) between the two groups. In addition, based on the results of multivariate analyses, EI of meropenem was considered as the predictive factor of the 3-day clinical effectiveness rate (*p* = 0.008) (Appendix A). EI of meropenem was not associated with 3-day microbial clearance in VLBW infants (*p* = 0.051). Nevertheless, gestational age (*p* = 0.013), hypotension (*p* = 0.029), or scleroderma (*p* < 0.001) were believed to be predictive factors (Appendix A).

#### 2.3.2. Two-Hour EI of Meropenem in Neonatal Sepsis

We performed a subgroup analysis in the EI group, including 67 patients that underwent a 2 h infusion and 60 patients that received a 3 h infusion of the same drug; the baseline characteristics of both groups were not statistically different (Appendix A). Univariate analysis results showed that neonates that underwent 3 h EI of meropenem had similar outcomes to those who underwent 2 h infusion in terms of 3-day clinical effectiveness rate (*p* = 0.325), microbial clearance rate (*p* > 0.999), CRP change (*p* = 0.445), and WBC recovery rate (*p* = 0.632) (Appendix A). Given that all of the *p*-values of outcomes in univariate analyses were more than 0.200, we did not conduct further multivariate analyses.

## 3. Discussion

Using real-world data of 256 children with neonatal sepsis, this study explored the pros and cons of EI and STI of meropenem. The results demonstrated that extending the infusion time of meropenem to approximately 2–3 h could improve clinical symptoms and the clearance of pathogenic bacteria in neonatal sepsis, with satisfactory tolerance and safety. There were no cases of recurrence in the EI or STI group after meropenem treatment. Previously, the effectiveness and safety of EI of meropenem were investigated in adult sepsis, ventilator-associated pneumonia, and severe infections [8,9,10]. In addition, Zhou et al. summarized clinical trials, case reports, and PK evidence on EI of meropenem in children with severe infection [11]. The authors concluded that although the PK theory was sufficient, the limited evidence did not support routine EI of meropenem in children, and further high-quality clinical control studies or observational studies needed to be conducted. To the best of our knowledge, to date, our historical cohort study has used the largest sample size to address EI of meropenem in neonatal sepsis, and our findings indicate the advantages of EI of meropenem in severe neonatal infections.

Neonatal sepsis is a systemic inflammatory response syndrome caused by bacterial, viral, or fungal infections. Currently, there is no consensus on the diagnosis of neonatal sepsis, and diverse definitions have been adopted by different research centers [12,13,14]. We considered that neonates often need empirical anti-infective treatment before etiological results are available, so neonates with the clinical diagnosis of sepsis were also included in the study. In addition, we did not distinguish between early-onset and late-onset sepsis based on age because meropenem is used in both diagnoses in China [15]. A total of 91 patients (35.55%) received empirical anti-infective treatment before using meropenem, which might have been initiated due to treatment failure or development of drug-resistant strains. There was no difference in the proportion of antibiotic initiation in the two groups, and pre-drug administration had no effect on the effectiveness of meropenem. In addition to the infusion duration, gestational age, hypotension, prolonged CRT, vomiting, and scleroderma were considered to affect the effectiveness outcomes, which might be related to the severity of the disease, thereby affecting the therapeutic success [16,17,18].

Meropenem is a broad-spectrum carbapenem antibacterial drug and shows strong antibacterial activity, high stability to β-lactamase, and a satisfactory safety profile [19]. It is a commonly used drug for the treatment of severe neonatal infections, mixed infections, drug-resistant bacteria, and enzyme-producing bacterial infections. However, considering the increasing number of enterobacteria of the extended β-lactamase spectrum and multi-drug-resistant pathogens, as well as the pathological changes in neonatal sepsis, it is vital to improve therapeutic effectiveness based on the PK/PD optimization theory. Optimum antibacterial activity is achieved when 40% fT  >  MIC (70% or even higher in severe infection) [20]. In certain situations, the aforementioned targets can be attained by increasing the frequency of administration, increasing the dosage, or extending the infusion duration [21]. However, in neonatal infections, it is inappropriate to further increase the dosing frequency and dosage strength owing to safety concerns; therefore, extending the infusion duration [22] is one of the ways to increase the probability of target attainment (PTA). Our study confirmed the advantages of EI of meropenem in reducing pathogenic bacterial load, improving clinical symptoms, and improving clinical effectiveness, with the confounding factors between the EI and STI groups having been appropriately controlled.

Although the advantages of admission into the neonatal intensive care unit (NICU) are obvious, in cases where the newborn does not require specialized activities in the unit, EI of meropenem seems to be an acceptable treatment choice for sepsis. However, considering that newborns are a special population group, EI of meropenem is still an off-label procedure which needs to be administered after receipt of informed consent. The stability of EI meropenem solution is also vital in such cases. Reportedly, 5.15 h after preparation, meropenem infusion solution degraded by 10% at 25 °C [23], and the degradation significantly accelerated with an increase in temperature. Therefore, it is not recommended to extend the infusion duration for too long (such as a continuous infusion), with the recommended period being approximately 2–3 h [24]. The solution for infusion should be prepared in the ward to avoid temperature and environmental fluctuations during transportation. In addition, meropenem infusion should be considered when severe sepsis or high drug resistance is suspected or confirmed. However, EI of meropenem tends to be used conservatively to reduce the risk of death related to sepsis, because the clinical manifestations of neonatal sepsis are not always specific, the disease progresses quickly, and the reporting of pathogenic results takes time. Neonates with meningitis were not included in this study mainly because the dose of meropenem (40 mg/kg/d) required for central nervous system infections is higher than that for neonatal sepsis [25]. A previous study [26] reported that EI of meropenem may reduce the PTA in cerebrospinal fluid; therefore, whether EI of meropenem in a newborn with meningitis is therapeutically appropriate remains inconclusive. Furthermore, the PK/PD advantages of EI meropenem are reportedly not observed in VLBW infants [27], which is inconsistent with our subgroup analysis results. Therefore, the clinical findings from this population require further research.

This study has several limitations. Owing to the retrospective study design and single-center data, the external validity of the findings may be reduced. We could not figure out the exact reasons why EI or STI administration was chosen initially, and we also failed to further evaluate the association between EI or STI route with death (20 cases) and complications (15 cases). Some of the patients (40 cases; 15 cases in the EI group, 25 cases in the STI group) recovered after 4- to 13-day treatment, but we did not further compare the effectiveness between EI and STI routes, considering the limited samples. Additionally, electronic records historically analyzed for safety are imperfect and probably biased, owing to difficulties in the judgment of drug-related or sepsis-related AEs in neonates and missing records of common AEs Another key limitation of this study was that subjective judgment according to medical records could not be entirely excluded although we defined a standard of 3-day clinical effectiveness based on various symptoms. Owing to the rapid progression and high mortality of neonatal sepsis, bacterial culture and drug susceptibility results may not be available when antibacterial therapy is initiated. Therefore, we failed to distinguish the appropriateness of EI of meropenem in various MIC reports. Furthermore, at present, EI of meropenem is not a routine regimen; therefore, the findings of this single-center cohort study need further verification in other medical centers.

## 4. Materials and Methods

### 4.1. Study Sample

A single-center historical cohort study was conducted to evaluate the effectiveness and safety outcomes of EI compared with STI of meropenem in neonatal sepsis using electronic health records from Peking University Third Hospital. We included all neonates diagnosed with sepsis and treated with meropenem in the NICU from 1 December 2011 to 1 April 2021. Previously, the proportion in the EI group was assumed to be 82% and 56.8% in the STI group [7]. We calculated that a sample of 98 patients (49 in the EI group; 49 in the STI group) would provide the study with 80% power to detect a difference between the group proportions of 25.2% at a two-sided alpha of 0.05. Given an anticipated dropout rate of 20%, the total sample size required was 123.

Under the premise of clinical abnormalities, sepsis diagnosis was defined as follows: when blood culture or other sterile cavity fluid culture tested positive for bacteria. In addition, when the culture results were negative, but the neonate received antibiotic treatment for no less than 5 days, without infection elsewhere, and met any of the following criteria: (1) no less than two positive blood non-specific tests; (2) cerebrospinal fluid test abnormalities; or (3) detection of DNA or antigens of specific bacteria in the blood. Neonates with Gram-positive cocci sepsis or purulent meningitis comorbidities were excluded. 

### 4.2. Exposure and Outcomes

The main exposure was the infusion duration of meropenem (20 mg/kg/dose), which was 0.5 h (STI group) and 2–3 h (EI group). Effectiveness outcome measures were the rate of 3-day clinical effectiveness and 3-day microbial clearance. Neonates were defined as having attained clinical effectiveness when they achieved normal body temperature (36.5 °C to 37.5 °C), blood pressure, and heart rate, stable hemodynamics, relief of dyspnea, and good intestinal tolerance, without the need for invasive mechanical ventilation [28,29,30] (Appendix A). Two experienced pediatricians, who were blinded to infusion duration, independently evaluated the effectiveness. Microbial clearance was determined based on the pathogenic culture results. Other laboratory abnormalities were defined as secondary outcome measures, including a 3-day recovery rate of CRP or WBC. Safety outcome measures were the rate of BUN (>7.5 mmol/L), creatinine (>130 μmol/L), or ALT (>70 μ/L) abnormalities in 3 days.

### 4.3. Data Analysis

Overall participant demographics and clinical characteristics were reported and stratified by the infusion duration. To determine potential confounders and matching variables, risk factors were identified as a priority to assess differences in demographic and clinical characteristics between EI and STI groups.

All analyses were performed with R version 4.0.3 (R Foundation for Statistical Computing, Vienna, Austria). Data distribution was determined according to P-P plot and Q-Q plot. If continuous variables conformed to a normal distribution, they were reported as the mean ± standard deviation (SD), and the independent samples *t*-test was conducted for comparison between two groups. If they did not conform to a normal distribution, the median (25% quantile, 75% quantile) was used, and the Wilcoxon test was applied to compare the two groups. The enumeration data were described by the number of cases and constituent ratios, and the χ^2^ test or Fisher’s exact test was used for comparison between groups. We identified the 3-day clinical effectiveness rate and the 3-day microbial clearance rate as dependent variables. Binary logistic regression was conducted to perform multivariate analysis to determine whether EI and STI were independent factors for outcomes, and to explore other influencing factors. Furthermore, we performed subgroup analyses in VLBW infants versus normal weight neonates, was well as 2 h versus 3 h EI of meropenem. All statistical tests were two-sided, and results with *p*  <  0.05 were considered statistically significant.

## 5. Conclusions

In conclusion, based on real-world data, EI of meropenem may be associated with better effectiveness and comparable safety in the treatment of neonatal sepsis than STI therapy. Neonates on meropenem therapy who are suspected of serious sepsis with related symptoms may be administered an empirical extended infusion. However, due to the historical study design and limited sample size, the safety evaluation might be biased, and these findings need to be further confirmed in high-quality randomized controlled trials of larger sample sizes.

## Figures and Tables

**Figure 1 antibiotics-11-00341-f001:**
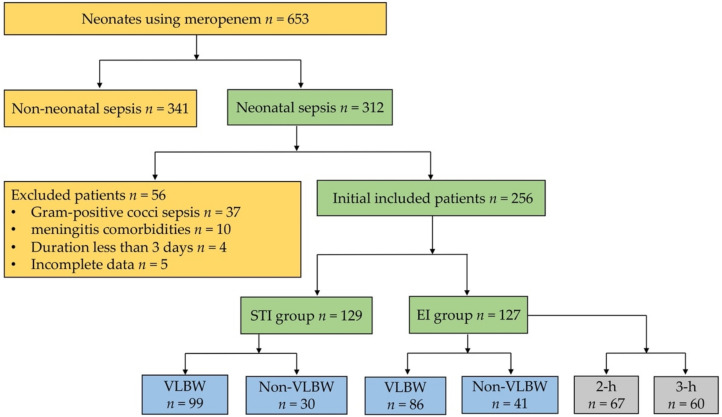
Inclusion and exclusion criteria of participants.

**Table 1 antibiotics-11-00341-t001:** Baseline information of the patients finally included in the study.

Characteristic	EI Group (*n* = 127)	STI Group (*n* = 129)	*p*-Values
Age (weeks), mean (SD)	31.1 (3.49)	30.5 (3.05)	0.166
Weight (grams), mean (SD)	1460 (722)	1340 (608)	0.228
Poor response	110 (86.6%)	103 (79.8%)	0.147
Antibiotics upgrade	39 (30.7%)	52 (40.3%)	0.109
CRT extension	11 (8.7%)	18 (14.0%)	0.182
Hypothermia	23 (18.1%)	15 (11.6%)	0.145
Apnea	60 (47.2%)	66 (51.2%)	0.531
Dyspnea	67 (52.8%)	71 (55.0%)	0.714
Abdominal distention	39 (30.7%)	31 (24.0%)	0.231
Vomiting	4 (3.1%)	5 (3.9%)	1.000
Heart rate increase	45 (35.4%)	43 (33.3%)	0.724
Hypotension	7 (5.5%)	10 (7.8%)	0.472
Cool extremities	12 (9.4%)	17 (13.2%)	0.347
Jaundice	19 (15.0%)	16 (12.4%)	0.551
Scleroderma	6 (4.7%)	4 (3.1%)	0.728
Pale or gray complexion	60 (47.2%)	51 (39.5%)	0.213
Convulsions	0 (0%)	3 (2.3%)	0.251

Note: EI, extended infusion; STI, short-term infusion; SD, standard deviation; CRT, capillary refill time.

**Table 2 antibiotics-11-00341-t002:** Univariate analysis results of EI of meropenem effectiveness outcomes.

Outcomes	EI Group (*n* = 127)	STI Group (*n* = 129)	*p*-Values
3-day clinical effectiveness rate	104 (81.9%)	77 (59.7%)	<0.001
3-day microbial clearance	120 (94.5%)	110 (85.3%)	0.015
3-day CRP recovery rate	30 (23.6%)	21 (16.3%)	0.141
3-day WBC recovery rate	67 (52.8%)	55 (42.6%)	0.105

Note: EI, extended infusion; STI, short-term infusion; CRP, C-reactive protein; WBC, white blood cell.

**Table 3 antibiotics-11-00341-t003:** Multivariate analysis results of EI of meropenem effectiveness outcomes.

Predictive Factors	Odds Ratio [95% CI]	*p*-Values	Odds Ratio [95% CI]	*p*-Values
**Outcomes**	**3-day clinical effectiveness rate**		**3-day microbial clearance**	
Infusion time	**0.34 (0.18,0.62)**	**0.001**	**4.13 (1.24, 13.78)**	**0.021**
Age	0.96 (0.82, 1.13)	0.628	**0.65 (0.47, 0.90)**	**0.010**
Weight	1.00 (1.00, 1.00)	0.227	1.00 (1.00, 1.00)	0.081
Poor response	0.47 (0.20, 1.13)	0.092	2.29 (0.39, 13.49)	0.361
Antibiotics upgrade	0.56 (0.29, 1.08)	0.082	0.45 (0.13, 1.54)	0.202
CRT extension	0.67 (0.20, 2.23)	0.512	**10.89 (1.50, 79.02)**	**0.018**
Hypothermia	1.36 (0.45, 4.09)	0.585	1.14 (0.16, 8.08)	0.894
Apnea	0.67 (0.36, 1.26)	0.219	0.56 (0.18, 1.73)	0.316
Dyspnea	0.69 (0.37, 1.29)	0.243	0.62 (0.18, 2.11)	0.447
Abdominal distention	2.06 (0.95, 4.46)	0.069	1.85 (0.49, 6.96)	0.361
Vomiting	0.42 (0.08, 2.27)	0.315	10.24 (1.15, 91.56)	0.037
Heart rate increase	0.67 (0.35, 1.29)	0.233	1.43 (0.45, 4.59)	0.543
Hypotension	4.75 (0.88, 25.62)	0.070	**5.79 (1.31, 25.58)**	**0.020**
Cool extremities	0.61 (0.21, 1.78)	0.364	1.06 (0.16, 7.14)	0.956
Jaundice	1.09 (0.42, 2.84)	0.856	0.32 (0.05, 2.26)	0.254
Scleroderma	0.28 (0.06, 1.25)	0.094	**40.15 (5.66, 284.83)**	**<0.001**
Pale or gray complexion	1.30 (0.68, 2.49)	0.424	0.87 (0.27, 2.8)	0.812
Convulsions	0.24 (0.02, 3.59)	0.300	0.68 (0.01, 35.67)	0.846

Note: EI, extended infusion; CRT, capillary refill time. The Bold Figures Indicate Statistical Differences.

**Table 4 antibiotics-11-00341-t004:** EI of meropenem safety outcomes.

Outcomes	EI Group (*n* = 127)	STI Group (*n* = 129)	*p*-Values
3-day BUN abnormality	50 (39.4%)	44 (34.1%)	0.383
3-day creatinine abnormality	49 (38.6%)	39 (30.2%)	0.160
3-day ALT abnormality	44 (34.6%)	42 (32.6%)	0.724

Note: EI, extended infusion; STI, short-term infusion; BUN, blood urea nitrogen; ALT, alanine transaminase.

## Data Availability

The data, including baseline information, processed data, and analyses in detail, are available from the authors upon reasonable request and with permission of corresponding authors.

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
