# Peer review of "Extended Infusion of Meropenem in Neonatal Sepsis: A Historical Cohort Study"

_antibiotics, 2022, doi:10.3390/antibiotics11030341_

Round 1
Reviewer 1 Report
Cao and colleagues conducted a very well study design and presentation to evaluate the efficacy and safety of using extended meropenem strategy in severe neonatal sepsis. In addition, the knowledge form the study is worth for general clinical practice. I have no corrections or comments.
Author Response
Dear reviewer,
Thank you so much for reviewing this manuscript, and we appreciated that you accepted our manuscript.
Best wishes,
Pengxiang Zhou on behalf of all co-authors
Reviewer 2 Report
- Manuscript needs to be revised to check for improper sentence formation, ex- line 46 would read better if it was like:- or attaining no-therapeutic success with maximum dose. There is an evident lack of logical flow in sentences.
2. In the beginning it should be clearly stated if this a data collection and analysis of an past dosing event or the researchers themselves planned a study and dosed 2 different set of patients.
3. If its an analysis of a past event, it would be great to shed light on why the practitioner choose to dose EI or STI in first place, whats the common norms to choose between EI and STI
4. Amount of dosage should have been discussed in the beginning, and background information on reported AUC that needs to be maintained for optimal clinical efficacy, should be discussed.
Author Response
Dear reviewer,
We appreciated that you spent precious time reviewing our manuscript and provided useful suggestions. We have revised the manuscript accordingly, and provide a point-by-point response here.
1. We revised this sentence in the Introduction Section, and added a transitional sentence before we introduced the EI and STI of meropenem.
2. We changed "retrospective cohort study" to "historical cohort study" in the whole manuscript to show reflect that this study was an analysis of a past event.
3. Owing to the retrospective study design, we could not figure out the exact reasons why EI or STI routes were chosen. To our best knowledge, no guidelines or clinical pathways give definite recommendations of this issue, which was also mentioned in the Introduction Section. Therefore, we considered it as a limitation and described it at the end of the Discussion Section.
4. The standard dosage of meropenem in neonatal sepsis is 20mg/kg/dose, and we have added the dosage in the Introduction Section and 4.2 Exposure and Outcomes. Also, we considered fT>MIC as the main measure to reflect the PK/PD optimization of meropenem (time-dependent antibiotics), so we did not mention AUC in the manuscript.
Thank you so much again for reviewing this manuscript, and we hope these responses and revisions will find you well. We will try our utmost to revise the manuscript for any further comments.
Best wishes,
Pengxiang Zhou, on behalf of all co-authors
Reviewer 3 Report
The following comments are related to the review of the manuscript entitled “Extended Infusion of Meropenem in Neonatal Sepsis: A Retrospective Cohort Study”. The treatment of the selected health problem is important in neonatal care.
First of all, I would like to make a methodological comment on the study design. In the manuscript title, as well as in the text, the authors declare that a retrospective cohort study was conducted. Although such a terminology is widely used, it is not the most appropriate since, by definition and nature, a cohort design is a prospective one (analysis goes prospectively from exposure to outcome). Therefore, the correct wording would be “historical” meaning that data was collected prior to the study begin and the time sequence was from meropenem infusion (short or extended) to health outcome (three-day clinical efficacy and three-day microbial clearance). Please, consider it.
In lines 60-61, since explanations on Methods appear later in the manuscript, the expression “which was much higher than 123” must be rewritten; it might be better to say “which was larger than 123, the estimated sample size (see 4.1. Study Sample)”.
The manuscript is well written and tables are clear in their layout. However, reducing the number of decimal positions to 2 in ORs estimations and their 95% CI in Table 3 and to 1 in the percentages in the supplementary materials (as it is done in the main text) would lighten the reading and understanding process. Moreover, a table foot explaining that bold figures indicate statistical differences should be included when required (Table 3 and STable 4).
The authors must acknowledge other limitations of their research. The observational character of the design (real-world data) and the one-single center condition deserve attention in the Discussion section. In fact, in the Conclusions, the need for randomized clinical trials and larger samples of patients is stated and we have to remind that the conclusions must be drawn from the results and subsequent discussion.
Author Response
Dear reviewer,
We appreciated that you spent precious time reviewing our manuscript and provided useful suggestions. We have revised the manuscript accordingly, and provide a point-by-point response here.
1. We agree with your comments, and we changed the "retrospective cohort study" to "historical cohort study" in the whole manuscript as a result.
2. We have revised this sentence according to your suggestion.
3. We have modified the number of decimal positions, and added the table foot explanation according to the comments.
4. The limitation of retrospective study design was added in the limitation section, so the conclusion could be well conducted subsequently.
Thank you so much again for reviewing this manuscript, and we hope these responses and revisions will find you well. We will try our utmost to revise the manuscript for any further comments.
Best wishes,
Pengxiang Zhou, on behalf of all co-authors